# Hippocampal neural stem cells facilitate access from circulation via apical cytoplasmic processes

Tamar Licht[1,2]*, Esther Sasson[1†], Batia Bell[1†], Myriam Grunewald[1], Saran Kumar[1], Tirzah Kreisel[3], Ayal Ben-Zvi[1], Eli Keshet[1]*

[1]Department of Developmental Biology and Cancer Research, Faculty of Medicine, the Hebrew University, Jerusalem, Israel; [2]Department of Medical Neurobiology, Faculty of Medicine, the Hebrew University, Jerusalem, Israel; [3]Edmond and Lily Safra Center for Brain Sciences (ELSC), The Hebrew University Givat Ram, Jerusalem, Israel

**Abstract** Blood vessels (BVs) are considered an integral component of neural stem cells (NSCs) niches. NSCs in the dentate gyrus (DG(have enigmatic elaborated apical cellular processes that are associated with BVs. Whether this contact serves as a mechanism for delivering circulating molecules is not known. Here we uncovered a previously unrecognized communication route allowing exclusive direct access of blood-borne substances to hippocampal NSCs. BBB-impermeable fluorescent tracer injected transcardially to mice is selectively uptaken by DG NSCs within a minute, via the vessel-associated apical processes. These processes, measured >30 nm in diameter, establish direct membrane-to-membrane contact with endothelial cells in specialized areas of irregular endothelial basement membrane and enriched with vesicular activity. Doxorubicin, a brain-impermeable chemotherapeutic agent, is also readily and selectively uptaken by NSCs and reduces their proliferation, which might explain its problematic anti-neurogenic or cognitive side-effect. The newly-discovered NSC-BV communication route explains how circulatory neurogenic mediators are 'sensed' by NSCs.

*For correspondence:
tamarli@ekmd.huji.ac.il (TL);
elik@ekmd.huji.ac.il (EK)

†These authors contributed equally to this work

Competing interests: The authors declare that no competing interests exist.

## Introduction

Neural stem cells (NSCs) serve as a source for new neurons in both the developing and adult brain. In the adult rodent brain, NSCs are confined into two locales, namely the subventricular zone (SVZ) of the lateral ventricles and the DG of the hippocampus. NSCs within the DG proliferate and differentiate into granule cells (GCs) eventually integrating into the existing network (*van Praag et al., 2002*). DG NSCs [also known as radial-glia-like cells (RGLs) (*Bonaguidi et al., 2011*) are distinguished by a unique, tree-like morphology with soma embedded in the subgranular zone (SGZ), a major shaft extending through the adjacent granule cell layer (GCL) and terminating in a dense network of fine cytoplasmic processes spreading in the inner molecular layer (*Gebara et al., 2016*). The functional role of these terminal shafts (if any) has remained obscure.

Stem cell functioning in general, and of RGLs in particular, relies on microenvironmental support by various cellular and extracellular components, collectively referred to as the 'stem cell niche'. Blood vessels (BVs) are considered an integral, indispensable component of stem cell niches, including NSC niches (*Licht and Keshet, 2015*; *Schildge et al., 2014*). BV may impact stem cell performance by two different modes: via locally released blood-borne substances or via paracrinically acting factors elaborated by the endothelium ('angiocrine factors')(*Rafii et al., 2016*).

Adult hippocampal RGLs are mostly quiescent but undergo several rounds of cell division upon activation (*Encinas et al., 2011*; *Pilz et al., 2018*). Neurogenic rates are greatly affected by systemic

cues, exemplified by exercise-enhanced neurogenesis shown to be mediated by systemic factors (*Fabel et al., 2003*; *Ma et al., 2017*; *Moon et al., 2016*). Likewise, heterochronic parabiosis experiments have demonstrated reciprocal influences on neurogenic rates mediated by systemic factors such as CCl11 or GDF11 (*Katsimpardi et al., 2014*; *Villeda et al., 2011*; *Villeda and Wyss-Coray, 2013*). Because hippocampal BVs (including those of the DG where RGLs reside) are thought to have a fully-functional BBB, the question arises how are circulating substances accessible to the brain parenchyma without a loss of BBB function?

Both SVZ-resident type B NSCs and DG-resident RGLs were shown to be intimately associated with BVs. In the SVZ, NSCs send a cellular process that wraps a capillary in the nearby striatum (*Lacar et al., 2012*; *Licht and Keshet, 2015*; *Mirzadeh et al., 2008*). In the DG, RGLs are physically associated with BVs at both the soma side (*Filippov et al., 2003*; *Licht and Keshet, 2015*; *Palmer et al., 2000*) and the apical side (*Licht and Keshet, 2015*; *Licht et al., 2016*; *Moss et al., 2016*). However, the functional significance of RGL-BV engagements is not well understood. Here we examined the proposition that these RGL processes might serve as a focal conduit through which blood-borne substances can be directly transferred to RGLs in a fully-functional BBB milieu, thereby providing a mechanistic explanation of how can NSCs 'sense' circulating neurogenic mediators. Additionally, we show that systemically-injected doxorubicin (a BBB-impermeable DNA-intercalating agent used for non-cerebral tumors) can be uptaken by RGLs and provide a direct anti-mitotic mechanism for its anti-neurogenic effect.

## Results

### RGL processes contact endothelial cells in basement membrane-reduced zones

The proposition that the thin processes extended by RGLs can serve as a bridge for transferring blood substances to RGL cell bodies requires that RGL-BV contact points are not separated by pericytes or by a basement membrane (BM) in which endothelial cells are usually invested and also constitutes an important component of the BBB (*Thomsen et al., 2017*). Previous studies that used Nestin-GFP to label RGLs (*Filippov et al., 2003*; *Moss et al., 2016*) were limited by the fact that mural pericytes are also labeled by this reporter (see *Figure 1a*; *Nakazato et al., 2017*), which makes perivascular pericytes and RGLs indistinguishable at the electron microscopy (EM) level. To overcome this shortcoming, we used *Gli1-cre^ERT2* mice [a mouse line in which inducible Cre is not expressed in DG cells other than RGLs (*Ahn and Joyner, 2004*) crossed to Ai9 (TdTomato) reporter line. RGLs were then highlighted with anti-RFP antibody and DAB staining (*Figure 1b*) and the tissue processed for EM imaging. Soma of DAB-labeled RGLs in the SGZ were readily discernible by their DAB dark grains (*Figure 1c*). At the inner molecular layer in the vicinity of those soma, we identified capillaries that are tightly wrapped by extra-fine (as thin as 30 nm in diameter) RGL processes labeled with DAB (*Figure 1d–g*).

The BM is visualized in EM images as a grayish 50–100 nm-thick layer surrounding the abluminal aspect of endothelial cells (*Figure 2a*). This component was found to thoroughly encircle DG capillaries of the hilus and outer molecular layer. Strikingly, however, observing endothelial-RGL processes contact points in the inner molecular layer revealed that engagement with BVs takes place in endothelial surfaces distinguished by irregular BM structures (absence of a classical BM appearance), thus allowing for possible direct membrane-to-membrane contact without the barrier imposed by the BM (*Figure 2b*). Based on TEM images, we quantified BM coverage at RGL contact points in comparison to areas devoid of RGL and found a significant difference (Fig, 2 c). Only ~30% of vessel abluminal aspect in contact with RGL retain the typical BM structure (as shown in *Figure 2c* right).

We further examined laminin (a vascular BM component)ith immunofluorescence using confocal microscopy (*Figure 3a–d*) followed by super-resolution microscopy (stochastic optical reconstruction microscopy – dSTORM, *Figure 3d'–m*). We analyzed images of Gli+ RFP-labeled RGLs in contact with blood vessels (*Figure 3c–i*) and found reduced laminin immunofluorescence where RGLs processes were present. In comparison, vessels in the DG not in contact with RGLs (*Figure 3j,k*) and cortical vessels (*Figure 3l,m*) had more pronounced laminin immunofluorescence. Quantification of dSTORM laminin signals (normalized to vessel wall length) revealed reduced laminin abundance at

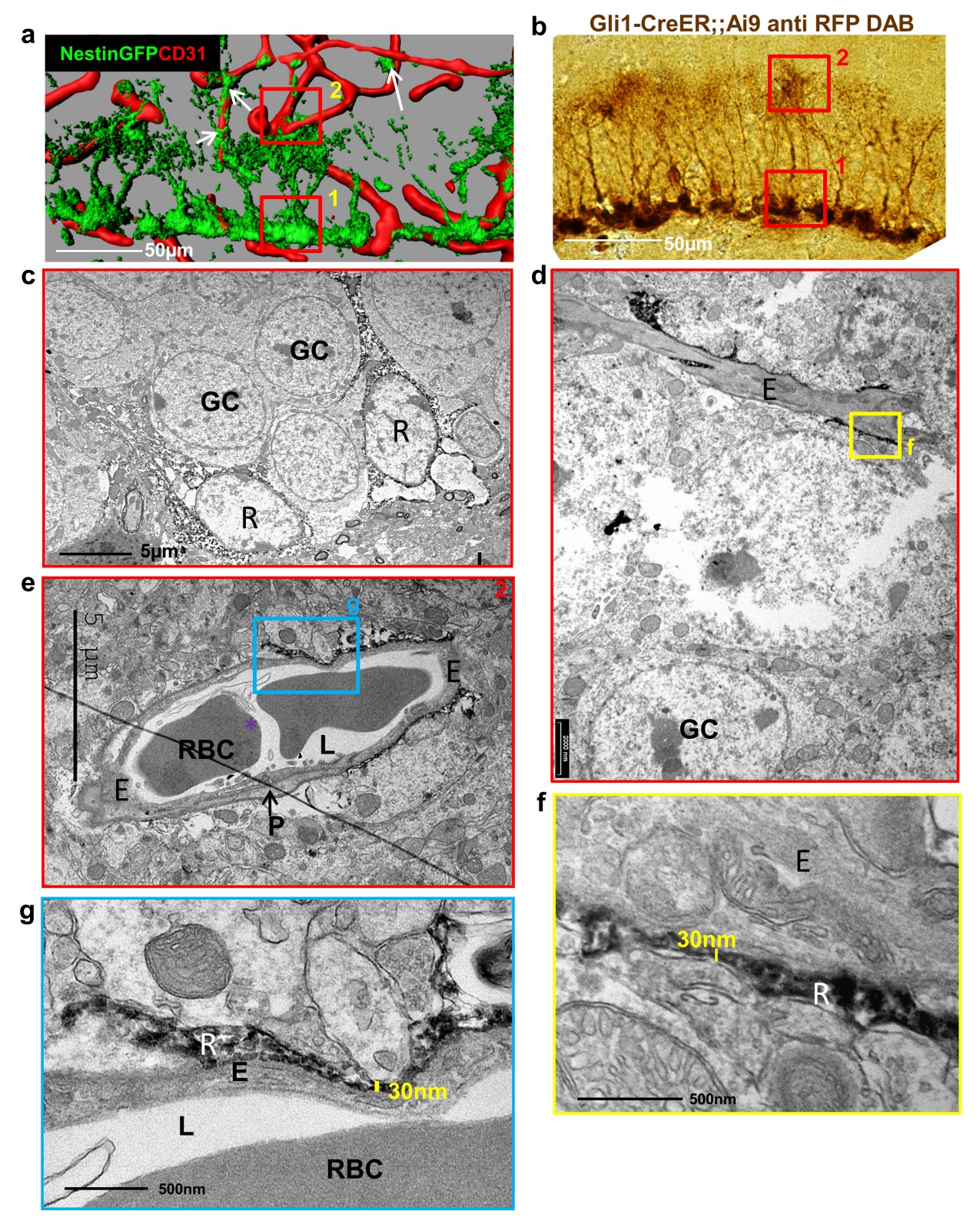

**Figure 1.** Fine apical processes of RGLs enfold capillary BVs in the inner molecular layer. (a) 3D reconstruction of the DG of Nestin-GFP mouse co-stained with CD31 for blood vessels. In this line, both RGLs and pericytes (arrows) are GFP-labeled. Please note the contact points of RGLs and blood vessels at both the basal side at the SGZ (1) and the apical side at the inner molecular layer (2). (b) *Gli1-Cre^{ERT2}* crossed to Ai9 line (inducible TdTomato reporter) received tamoxifen three days before brain retrieval. Immunohistochemistry was done using anti-RFP antibody and DAB labeling. Note that

*Figure 1 continued on next page*

*Figure 1 continued*

only RGLs are labeled in the DG of this mouse line. (1) basal side, (2) apical side. Scale bar, 50 μm. (c) Transmission electron microscopy (TEM) images taken at the SGZ (basal side) showing the soma of two DAB-labeled RGLs (R). The cytoplasm of those cells is detected by DAB dark grains. GC-granule cells. (d and e) TEM images of the inner molecular layer (apical side) showing two representative examples of a capillary wrapped with DAB-labeled RGL processes. (f and g) Higher magnification of (d) and (e) showing that RGLs processes can reach up to 30 nm in thickness. R-RGL; GC-granule cell; E-endothelial cell, RBC-red blood cell; L-blood vessel lumen; P-pericyte.

vessels which included a contact point with RGL in comparison with DG vessels without RGL process or in comparison with cortical vessels (*Figure 3n*).

Immunofluorescence for a second major protein of the BM, namely collagen IV, also revealed an apparent reduction in its expression in areas of RGL-BV contact (*Figure 3—figure supplement 1*).

## Vesicular activity at the RGL-BV contact points

The BV-RGL interface was also characterized by robust vesicular activity as many cytoplasmic and abluminal membrane-bound vesicular-like structures (measured 60–100 nm in diameter) were detected there (*Figure 4a–c* and *Figure 4—figure supplement 1a*). To highlight the nature of vesicular structures and to verify that these are transcytotic vesicles, we indtrouced HRP into the blood circulation and stain brain slices with DAB to label the transcytotic vesicular trafficking path. TEM images of capillaries at the inner molecular layer revealed luminal, cytoplasmic and abluminal vesicles containing HRP (*Figure 4d,e*). We next aimed to quantify the numbers of HRP-containing vesicles near an RGL. For this purpose, we injected HRP into the tail vein of a Gli1+ RFP-labeled mouse, and performed DAB histochemistry for both structures. Although this double-labeling procedure impaired the integrity of tissue and washed HRP from the vessel lumen, we were able to image and quantify HRP+ endothelial vesicles near a DAB-labeled RGL (*Figure 4—figure supplement 1b,d*). The density of these vesicles (vesicles/length of the abluminal boundary (μm)) at the BM-modified RGL-BV interface was greater in comparison with BV segments which did not contain an RGL as well as with BV-RGL interface separated by a clear BM (*Figure 4—figure supplement 1c,d*).

In view of these findings, this specialized neurovascular unit (NVU) was further tested for allowing the infiltration of additional systemically-derived molecules.

## BBB-impermeable circulatory molecular tracer is uptaken exclusively by hippocampal RGLs

Circumventricular organs are known to have permeable vessels to allow for direct exposure of neurons to circulating components. To examine the possibility that other cells in the intact adult brain are 'privileged' in the sense of having access to blood-borne substances, we injected into the systemic circulation a fluorescently-tagged 10Kd dextran tracer previously shown to be fully contained in the lumen of blood vessels with a functional BBB (*Ben-Zvi et al., 2014*; *Licht et al., 2015*). The tracer (6 mg/kg) was injected to the left ventricle of the heart (the tracer is rapidly cleared in the urine when injected to the tail vein) and the brain was retrieved one minute later and immediately fixed. Inspection of cryosections, co-stained with CD31 to highlight BVs, confirmed that - with the exception of the median eminence of the hypothalamus (a circumventricular organ) and choroid plexus vessels which are known to lack a functional BBB - the tracer is indeed fully retained in nearly all other cerebral BVs (*Figure 5a*). As expected, a small population of cells in the DG was labeled (*Figure 5a*, top left, and *Figure 5b*). These cells had their soma located in the SGZ, i.e., coinciding with the region where RGLs soma reside, and a major shaft extended towards the molecular layer. In fact, both the location and unique morphology of tracer-labeled cells strongly suggested that these are DG RGLs.

We wished to determine whether these cells serve as a conduit allowing direct transfer from the circulation without extracellular leakage into the parenchymal space. We followed the trajectory of tagged cells from a vessel-associated cellular process to soma. Considering its tortuous path, we used Imaris-aided 3D-reconstructions of serial confocal images. As shown in *Figure 5c*, there is a clear continuum in tracer presence from the vessel through the major shaft to the soma with no apparent signs of extracellular spillage.

To verify the identity of the tracer-accumulating cells, TRITC-labeled 10Kd tracer was injected into the circulation of mice harboring a Nestin-GFP reporter transgene highlighting RGLs (here, in

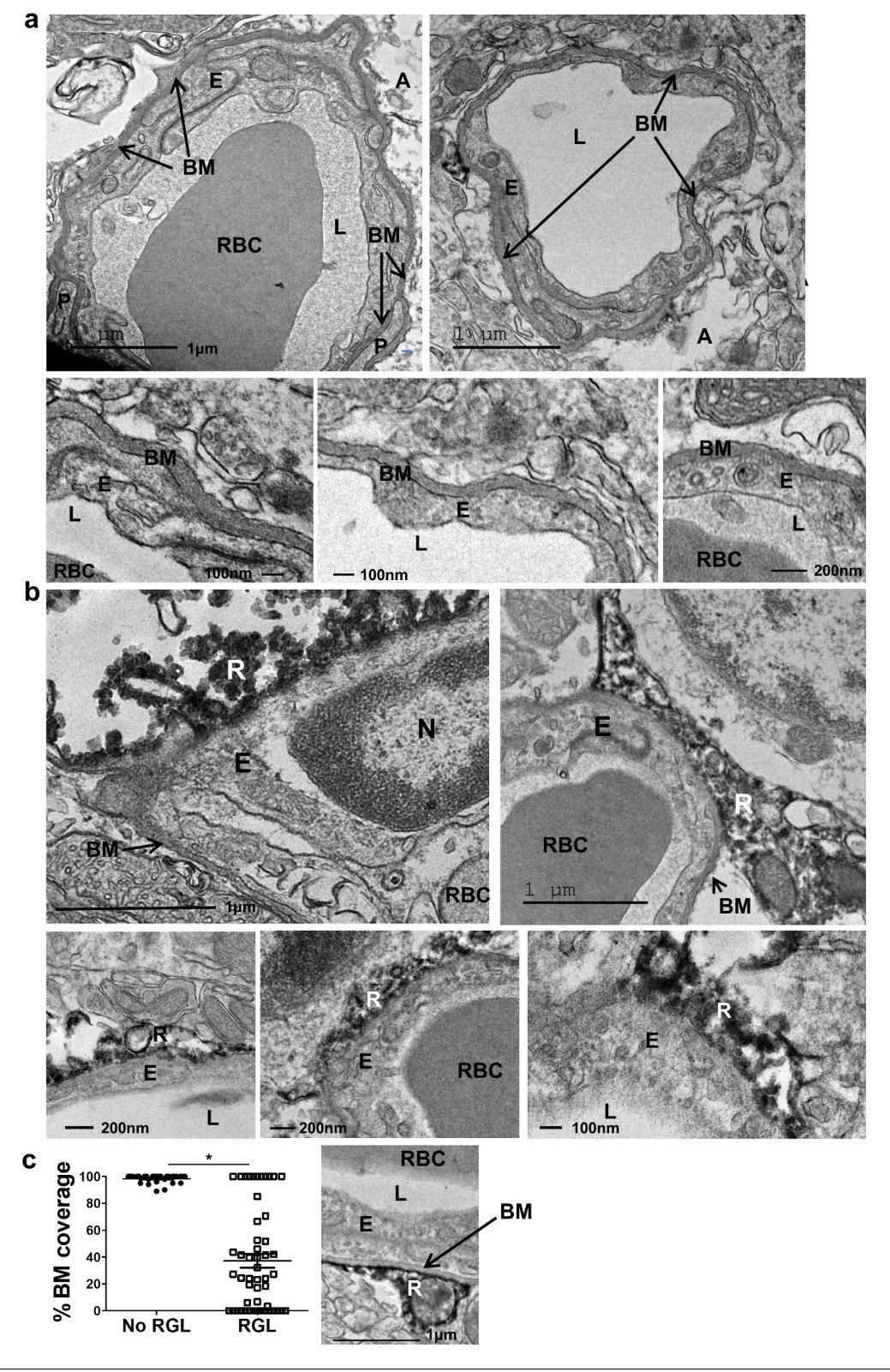

**Figure 2.** Absence of distinctive basement membrane at the contact points between RGLs and endothelial cells in the inner molecular layer. TEM images of the same samples as in *Figure 1*. (a) Representative images of capillaries from the hilus area. Note BM covering the entire capillary abluminal membrane. Please note a double-layered BM at the pericyte (P) contact point and a clear BM at the contact point with an astrocyte (A). Bottom– higher magnifications. Scale bars: top - 1 μm, bottom- 100 nm. (b) Representative images of capillaries taken from the inner molecular layer that associate with

*Figure 2 continued on next page*

*Figure 2 continued*

DAB-labeled RGLs. Note no characteristic BM at the contact point of RGL (R) with the endothelial cell (E). (**c**) Quantification for the % of BM-covered endothelial abluminal surface with or without contact with RGL. N = 2 animals, 55 images. t(53)=11.51. p=2.37*10$^{-5}$. Right: Image of BV-RGL contact with evident BM. R-RGL; E-endothelial cell; A-astrocyte; P-pericyte; RBC-red blood cell; L-blood vessel lumen; BM-basement membrane, N-endothelial nucleus.

contrast with EM studies, they can be distinguished from pericytes by their tree-like morphology). As shown in *Figure 6a*, due to their fineness, the tracer was barely visible in the tree-like processes and most of it accumulated in the major shaft and soma. We wished to re-confirm that RGLs thin processes also possess the tracer. For this purpose, we injected Nestin-GFP mice with biotin-labeled 10Kd dextran and visualized tracer distribution with Cy3-labeled streptavidin (this procedure was preferred to reduce interference of TRITC signal with GFP fluorescence). Co-localization of biotinylated tracer with nestin-GFP RGLs was evident in both the soma and the apical processes (*Figure 6b*).

## Direct uptake of the chemotherapeutic agent doxorubicin by RGLs explains its anti-neurogenic side effects

Doxorubicin (dox) is a widely-used chemotherapeutic agent of the Anthracyclines family of DNA - intercalating agents. While inaccessibility of this small molecule (543 Da) to the brain parenchyma (*Ohnishi et al., 1995*; *Treat et al., 2007*) has precluded its use for treating brain tumors, its use against other tumors is often associated with significant cognitive impairment, also known as 'chemo brain' (reviewed by *El-Agamy et al., 2019*). Findings in rodents showing that dox have a dramatic negative effect on DG neurogenesis (*Christie et al., 2012*; *Janelsins et al., 2010*; *Kitamura et al., 2015*; *Park et al., 2018*) prompted us to examine whether this seemingly paradoxical situation could be explained by direct selective uptake of systemically-administered dox by RGLs. To this end, an experimental protocol similar to the one used for following tracer trafficking was used, except that dox (6 mg/kg) replaced dextran injection. Here we took advantage of the fact that dox have the same fluorescent properties as commonly used red fluorophores thereby enabling its direct visualization in cell nuclei in PFA-fixed brain sections. The fact that systemically-injected dox does not cross to brain parenchyma was validated by showing that in all brain regions known to have a functional BBB, dox was detected only in endothelial cell nuclei (exemplified for the cortex and CA1 region of the hippocampus in *Figure 7a*), whereas in areas in which the BBB is inactive, dox-labeled nuclei were also readily detected outside the vasculature (shown for the choroid plexus and the median eminence of the hypothalamus in *Figure 7a*). An outstanding exception was dox uptake by RGLs, evidenced by significant labeling of RGL nuclei within 1–2 min from its injection into the circulation of nestin-GFP mice where co-labeling with GFP confirmed the RGL identity of dox-uptaking cells (*Figure 7b and c*). We next examined whether dox (being an anti-mitotic agent) has a negative effect on RGL proliferation. For this purpose, we injected dox at the dose of 5 mg/kg into the peritoneal cavity of Nestin-GFP mice (three injections, every other day) followed by three injections of BrdU (50 mg/kg, every 12 hr) to tag proliferating cells (*Figure 8a*). Notably, this chronic dox application (in comparison with the previously-used intracardial injection) does not allow for its detection in cell nuclei. Dox treatment did not have an effect on the total numbers of Nestin+ RGL population (*Figure 8b*, right), as the majority of those cells are not proliferating. We immunostained for both the proliferation marker Ki67 (*Figure 8b*) and for BrdU (*Figure 8c*) to detect the proliferating RGL populations. In both cases, the numbers of proliferating RGLs have decreased significantly following dox treatment (*Figure 8b and c*). We, therefore, suggest a direct antimitotic effect of dox as a presumable explanation for reduced neurogenesis reported in rodents treated with dox and as a possible factor in the 'chemo-brain' side effect of cancer patients.

## Discussion

Findings reported here showing that RGL-type NSCs in the adult hippocampus have a 'privileged' direct access to circulating blood molecules provide a possible mechanistic explanation to the enigmatic situation of NSCs 'sensing' the presence of particular blood components and responding by adjusting neurogenic rate despite the existence of a fully functional BBB milieu. Circulatory

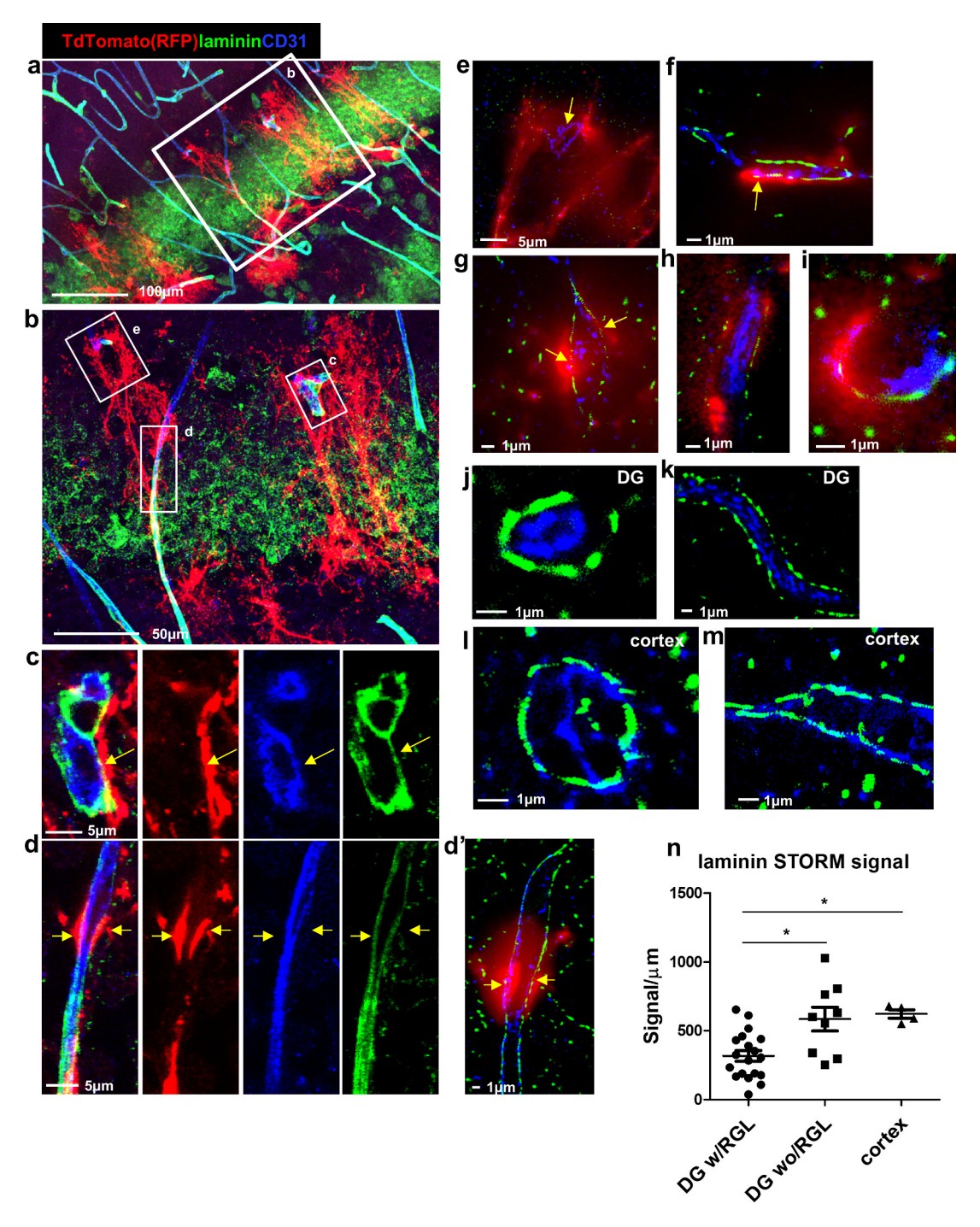

**Figure 3.** Reduced laminin expression at the BV-RGL contact point. Brain slice of *Gli1-Cre*[ERT2] mouse crossed to Ai9 reporter was immunostained with RFP, CD31 and pan-laminin antibodies. (a,b) Confocal image of the DG. (c,d) insets of b highlighting BV-RGL contact points. Arrows indicate areas of contact. (d') Super-resolution image of d. The red (568 nm) channel was kept in epi-flaurescence mode to highlight areas of BV-RGL contact. (e–i) Representative super-resolution images of capillaries in contact with RGL (arrows). (J,k) Representative images of capillaries in the DG not accosicated with an RGL. (l,m) Cortical capillaries. (n,) Quantification of laminin signal (in dSTORM images) in blood vessels of the DG (with and without a contact with RGL) and in the cortex, normalized to the length of the blood vessel wall (in μm). F(2,29) = 8.221, p=0.001. Post hoc analysis: DG w/RGL vs. DG wo/ RGL: p=0.005. DG w/RGL vs cortex: p=0.019.

The online version of this article includes the following figure supplement(s) for figure 3:

**Figure supplement 1.** Reduced Collagen IV immunohistochemistry at the RGL-BV interface.

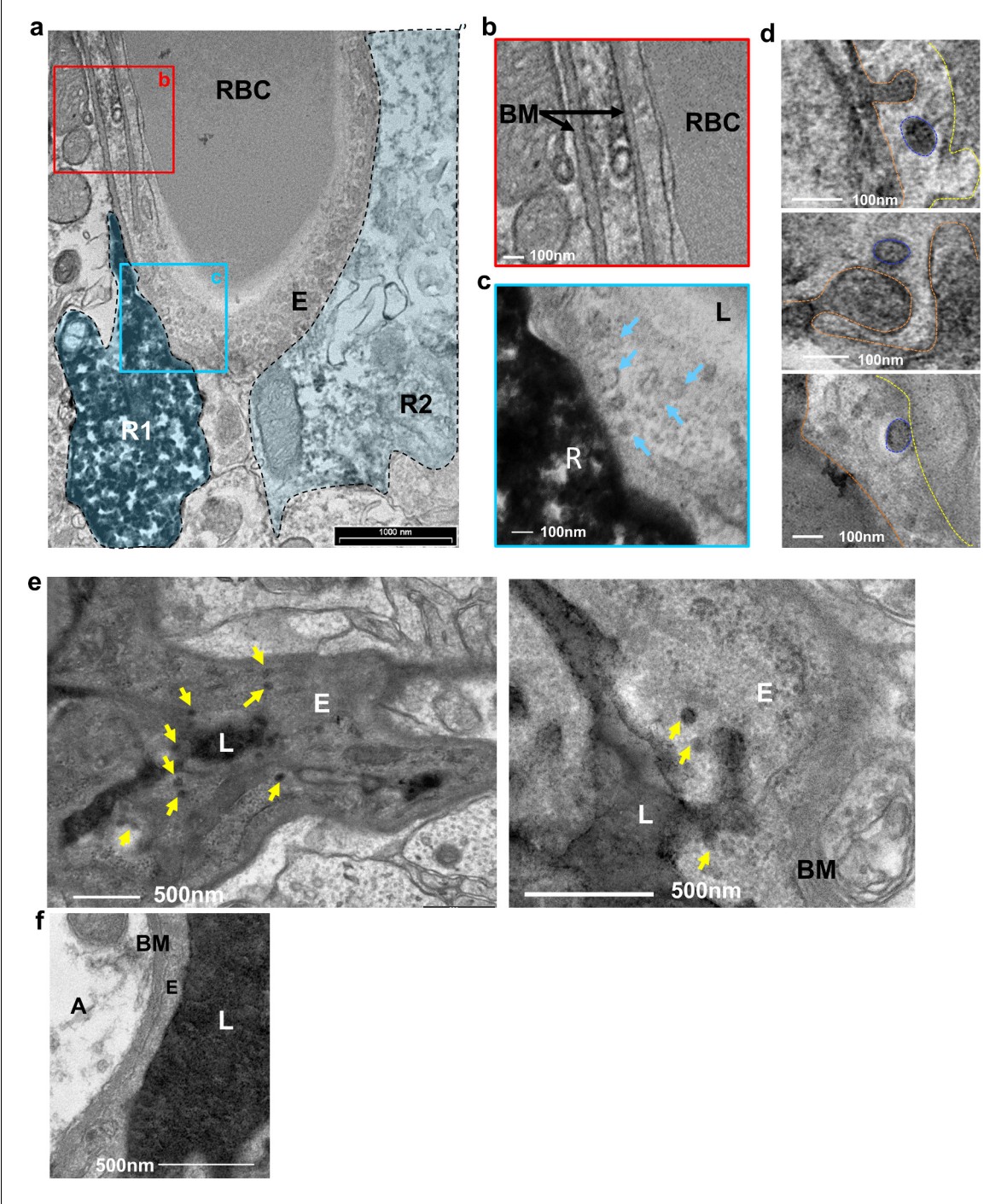

**Figure 4.** Vesicular activity at the BV-RGL interface. (a–c) TEM image of a BV in the inner molecular layer which was stained with anti-RFP antibody as in *Figure 1*, to highlight RGLs (R1, R2). BV-RGL contacts were examined for the presence of vesicular-like structures (50–100 nm circular structures at the cytoplasm, luminal and abluminal surfaces). Arrows highlight abluminal membrane-connected and cytoplasmic endothelial vesicles. (d) HRP was injected i.v 30 min before brain disection. Representative examples for luminal, cytoplasmic and abluminal HRP-filled vesicles are shown, luminal aspect (dashed orange line), vesicle (blue line) and abluminal boundary (yellow line). Images were taken from endothelial cells at the inner molecular layer. (e) TEM of endothelial cells at the inner molecular layer, injected with systemic HRP. Arrows highlight multiple vesicles containing HRP. (f) Representative image of an endothelial cell aspect with a continuous BM and no vesicular activity. E-endothelial cell, R-RGL, L-BV lumen, RBC-red blood cell, BM-basement membrane, A- astrocyte.

The online version of this article includes the following figure supplement(s) for figure 4:

**Figure supplement 1.** Vesicular activity in the BV-RGL contact point.

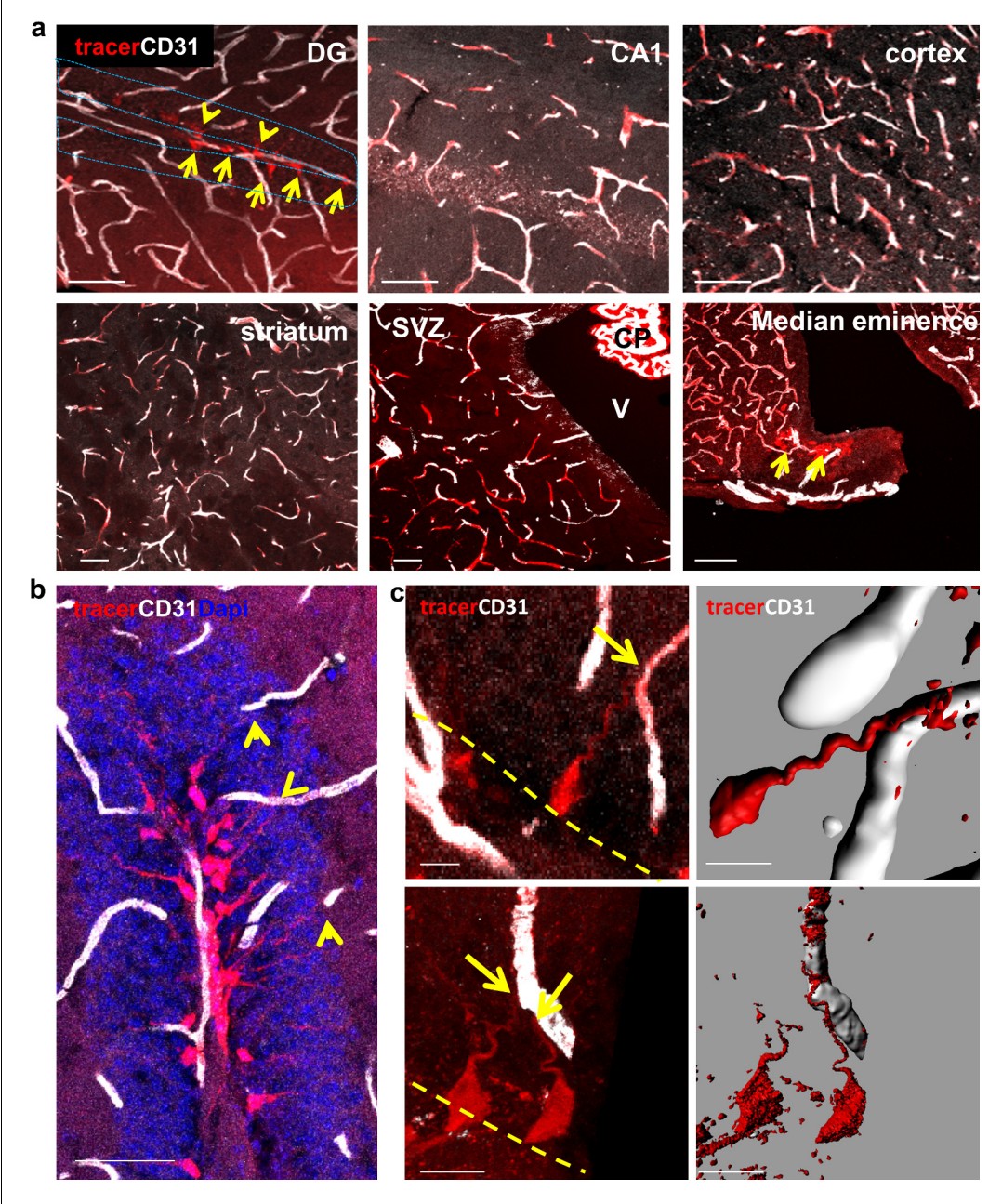

**Figure 5.** A unique population of cells in the SGZ is specifically labeled by systemically-injected 10Kd dextran tracer. (a) 10Kd dextran TRITC-labeled tracer was injected to the left heart ventricle of an anesthetized mouse. One minute later the brain was retrieved and immediately fixed. Sections (co-stained with CD31) of the indicated brain areas are presented. Tracer did not infiltrate into cells outside of the vasculature except for a specific population of cells in the SGZ of the DG (arrows), the CP and the median eminence of the hypothalamus (arrow). Scale bars, 50 µm. V-cerebral ventricle. CP – choroid plexus. (b) Representative high magnification image of the DG area in TRITC- 10Kd dextran-injected animal co-immunostained with CD31. Labeled SGZ cells have a radial process that is associated with local blood vessels in the inner molecular layer (arrows). Scale bar, 50 µm. (c) Higher magnifications and 3D reconstruction of a dextran-labeled cell and a capillary illustrating mutual contact with no labeling of the extracellular space. The yellow dashed line represents the SGZ. Scale bar, 5 µm.

molecules were shown to increase (or decrease) neurogenesis include specific cytokines whose circulatory levels change with age, as well as blood components accumulating in plasma following physical exercise and shown to upregulate DG neurogenesis (*Castellano et al., 2017*; *Cooper et al., 2018*; *Fabel et al., 2003*; *Leiter et al., 2019*; *Ozek et al., 2018*; *Smith et al., 2015*; *Villeda et al.,*

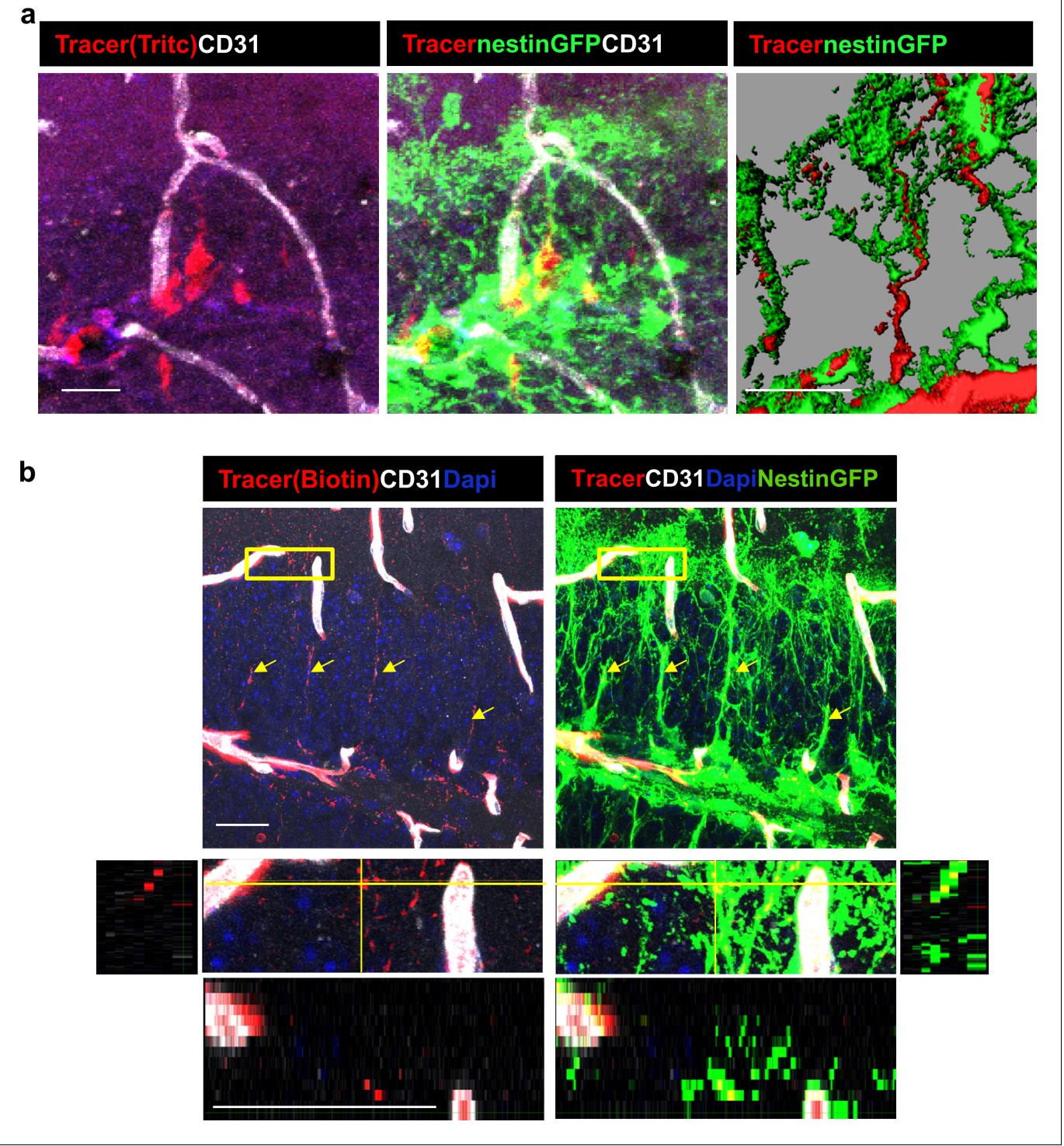

**Figure 6.** DG RGLs uptake a 10Kd systemically-injected dextran tracer. (a) TRITC-labeled 10Kd tracer was injected into a Nestin-GFP mouse. 3D Imaris reconstruction (right) and confocal z-plane (left, middle) images showing accumulation of the tracer in the soma and major shaft of RGLs. Scale bar, 20 µM. (b) Biotin-labeled 10Kd tracer was injected as in (a). Sections were stained for CD31 and tracer was highlighted by cy3-labeled streptavidin. Note tracer distribution throughout all RGL processes and soma (arrows), although less strong than the fluorescent tracer. The inset demonstrates the localization of tracer within GFP+ RGL fine apical processes. Scale bar, 20 µM.

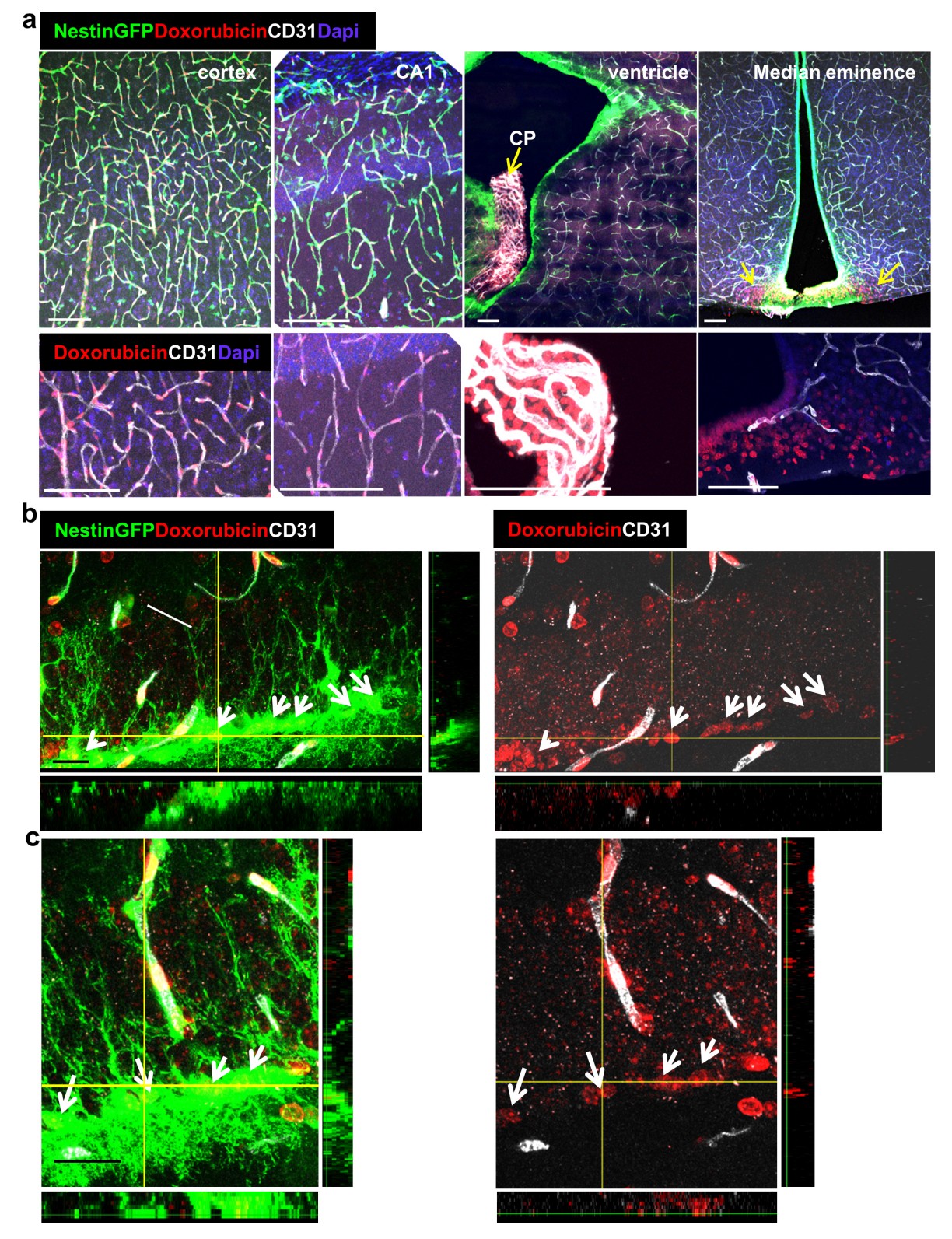

**Figure 7.** Doxorubicin has limited permeability through BBB capillaries but is being uptaken by RGLs. Doxorubicin (200 μg) was injected into the left heart ventricle of Nestin-GFP mice. The brain was retrieved two minutes later. Doxorubicin is integrated into the DNA and detected by self red spectrum fluorescence. (a) Immunofluorescence images co-stained with CD31 showing integration of doxorubicin to the nuclei of various brain areas. In areas with intact BBB such as the cortex and the CA1, doxorubicin integrates solely to capillary nuclei. In areas with non-barriered endothelium such as

*Figure 7 continued on next page*

*Figure 7 continued*

the choroid plexus and the median eminence, doxorubicin is detected externally to blood vessels (arrows). Scale bar, 100 μM. (**b** and **c**) Two representative high-magnification images of the DG demonstrating the integration of doxorubicin to the nuclei of GFP⁺ RGLs. Z-plane projections are shown. Scale bar, 20 μM.

*2014*). It is likely that additional physiologic cues impacting DG neurogenesis are also mediated by systemic factors making use of this direct gateway to DG RGLs.

Our finding showing that the transfer of blood-borne molecules to RGL cell bodies takes place through RGL's thin cytoplasmic processes ascribes a first known function to the obscure terminal arbor of RGLs. Noteworthy, unlike RGLs, SVZ NSCs failed to uptake injected molecules (*Figure 5a*) despite the fact that they also contact BV via a single extended process (*Mirzadeh et al., 2008*). Previously, we and others have shown that cytoplasmic processes extended by RGLs make direct contacts with nearby BVs (*Licht and Keshet, 2015*; *Licht et al., 2016*; *Moss et al., 2016*). Moreover, we showed that VEGF-promoted BV rejuvenation leads to remodeling of the terminal RGL arbor and engagement with even more remote capillaries (*Licht et al., 2016*). Yet, the functional significance of these physical RGL-BV associations has remained unknown until now.

We show, using EM imaging, that the fine processes of RGLs in the DG make direct contact with the membrane of nearby endothelial cell in the inner molecular layer. This contact takes place in areas with modified/reduced endothelial BM, as shown by both EM imaging and immunofluorescence/super-resolution microscopy. We also highlighted endothelial vesicular activity at the engagement points. While BBB vasculature is known to be poor in vesicular activity (*Ben-Zvi et al., 2014*), these findings suggest that this unique connection may allow for systemic molecules to be delivered via vesicular transcytosis directly to the membrane of RGLs. Further studies are required to verify how the transfer of molecules takes place and whether this process is discriminative with regards to specific molecules.

Indiscriminative uptake may explain apparent RGL vulnerabilities to otherwise BBB-impermeable molecules. A notable, clinically-relevant example is that of the BBB-impermeable chemotherapeutic agent doxorubicin. A major side effect of dox is a debilitating cognitive decline affecting a considerable fraction of treated cancer patients (*El-Agamy et al., 2019*) and animal studies have shown that treatment with dox is manifested in impeded neurogenesis (*Christie et al., 2012*; *Janelsins et al., 2010*; *Kitamura et al., 2015*; *Rendeiro et al., 2016*). However, it has been difficult to reconcile the adverse cerebral side effects of dox with the fact that it is a substrate for p-glycoprotein pumps and is not available to the brain. Our finding that systemically-injected dox is readily and efficiently uptaken by DG RGLs provides a more compelling explanation to this seeming paradox than previous suggestions arguing an indirect mechanism mediated by secondary molecules (*Janelsins et al., 2010*; *Kitamura et al., 2015*). We show here that the treatment by dox decreases proliferating RGL numbers thus suggesting a cell-autonomous anti-mitotic mechanism.

We conclude that this unique NVU pathway may serve as a gateway for molecules that need to bypass the barrier or a 'hole in the fence' allowing for penetration of molecules that should not enter the brain. We believe that the new mode of delivery of systemic molecules to RGLs uncovered by this study, not only provides a mechanistic explanation to systemic influences on adult hippocampal neurogenesis but may also provide new means of interventional modulations of this important process.

## Materials and methods

**Key resources table**

| Reagent type (species) or resource | Designation | Source or reference | Identifiers | Additional information |
|---|---|---|---|---|
| Strain, strain background (*Mus musculus* C57Bl/6) | *Nestin-GFP* | Prof. Grigori Enikolopov, CSHL andStonybrook | | |

*Continued on next page*

Continued

| Reagent type (species) or resource | Designation | Source or reference | Identifiers | Additional information |
|---|---|---|---|---|
| Strain, strain background (*Mus musculus* C57Bl/6) | *Gli1-Cre*^ERT2 | Jax mice | strain 007913 | |
| Strain, strain background (*Mus musculus* C57Bl/6) | Ai9 | Jax mice | strain 007909 | |
| Antibody | Mouse monoclonal anti BrdU | Serotec | PRID:AB_323427 | 1:400 |
| Antibody | Rat monoclonal anti-CD31 | Becton Dickinson | PRID:AB_393571 | 1:50 |
| Antibody | Rabbit polyclonal anti Laminin | Thermo Fischer Scientific | RRID:AB_60396 | 1:200 |
| Antibody | Rabbit polyclonal anti Collagen-IV | Abcam | PRID:AB_305584 | 1:200 |
| Antibody | Rabbit polyclonal Anti Ki67 | Thermo Fischer Scientific | PRID:AB_10979488 | 1:200 |
| Antibody | Rabbit polyclonal anti-GFP | Thermo Fischer Scientific | PRID:AB_221570 | 1:200 |
| Antibody | Rabbit polyclonal anti-RFP | Abcam | PRID:AB_945213 | 1:400 |
| Antibody | HRP Anti-Rabbit IgG | Vector | GZ-93951–40 | 1 drop |
| Antibody | Chick polyclonal anti RFP | EnCor | PRID:AB_2572308 | 1:300 |
| Antibody | Donkey anti-mouse Alexa647 | Jackson Immunoresearch | PRID:AB_234086 | 1:400 |
| Antibody | Donkey anti-rat Alexa647 | Jackson Immunoresearch | PRID:AB_2340694 | 1:400 |
| Antibody | Donkey anti-rabbit Alexa647 | Jackson Immunoresearch | PRID:AB_2492288 | 1:400 |
| Antibody | Donkey anti-rabbit Cy3 | Jackson Immunoresearch | PRID:AB_2307443 | 1:400 |
| Antibody | Donkey anti-rabbit Alexa488 | Jackson Immunoresearch | PRID:AB_2340619 | 1:400 |
| Antibody | Goat anti-hamster Alexa647 | Thermo Fischer Scientific | PRID:AB_2535868 | 1:400 |
| Antibody | Hamster polyclonal anti CD31 | Bio-Rad | PRID:AB_321653 | 1:50 |
| Commercial assay or kit | Streptavidin Alexa594 | Jackson Immunoresearch | PRID:AB_2337250 | 1:400 |

*Continued on next page*

*Continued*

| Reagent type (species) or resource | Designation | Source or reference | Identifiers | Additional information |
|---|---|---|---|---|
| Commercial assay or kit | Dab Plus | Abcam | ab103723 | |
| Chemical compound, drug | Florescent mounting medium | Thermo Fischer Scientific | TA-030-FM | |
| Chemical compound, drug | DAPI | Sigma | D9542 | |
| Chemical compound, drug | Tamoxifen | Sigma | T5648 | 8 mg/mouse PO |
| Chemical compound, drug | BrdU | Sigma | B5002 | 50 mg/Kg IP |
| Chemical compound, drug | Doxorubicin | Sigma | 44583 | 6 mg/kg IC 5 mg/kg IP |
| Chemical compound, drug | HRP | Sigma | P8250 | 10 mg/mouse IV |
| Chemical compound, drug | 10Kd Dextran, TRITC | Molecular Probes | D1817 | 6 mg/Kg IC |
| Chemical compound, drug | 10Kd Dextran, Biotin | Molecular Probes | D1956 | 6 mg/Kg IC |
| Chemical compound, drug | GLOX | Sigma | G2133 | 0.5 mg/ml |
| Chemical compound, drug | catalase | Sigma | C30 | 40 µg/ml |
| Chemical compound, drug | cysteamine MEA | Sigma | M9768 | 10 mM |
| Software, algorithm | ImageJ | NIH | Thunder STORM plugin | |
| Software, algorithm | SPSS | IBM | | |

## Mice

All animal procedures were approved by the animal care and use committee of the Hebrew University. Transgenic mouse lines that were used in this study (all of C57Bl/6 background): Ai9 and *Gli1-cre^{ERT2}* lines were purchased from the Jackson Laboratories (strains 007909, 007913). Nestin-GFP line was obtained from Prof. Grigori Enikolopov, CSHL (*Mignone et al., 2004*). Both males and females aged 2–3 months were used. Animals were grown in SPF housing conditions (4–6 animals per cage) with irradiated rodent food and water ad libitum and 12 hr light/dark cycle. Tamoxifen (Sigma T5648, 40 mg/ml in sunflower seed oil) was administered orally once daily for 2–3 days at a dose of ~8 mg/animal. BrdU (Sigma B5002) was dissolved in saline and injected I.P at 50 mg\kg every 12 hr (3 injections).

## Tracer and doxorubicin injection

The following dextran tracers were used (all are lysine fixable): Tritc-labeled 10kD dextran (Molecular Probes, D1817), biotin-labeled 10Kd dextran (D1956). Doxorubicin was obtained from Sigma-Aldrich (44583). All were dissolved in saline to 2 mg/ml and 100 µl (6 mg/kg) was injected into the left heart

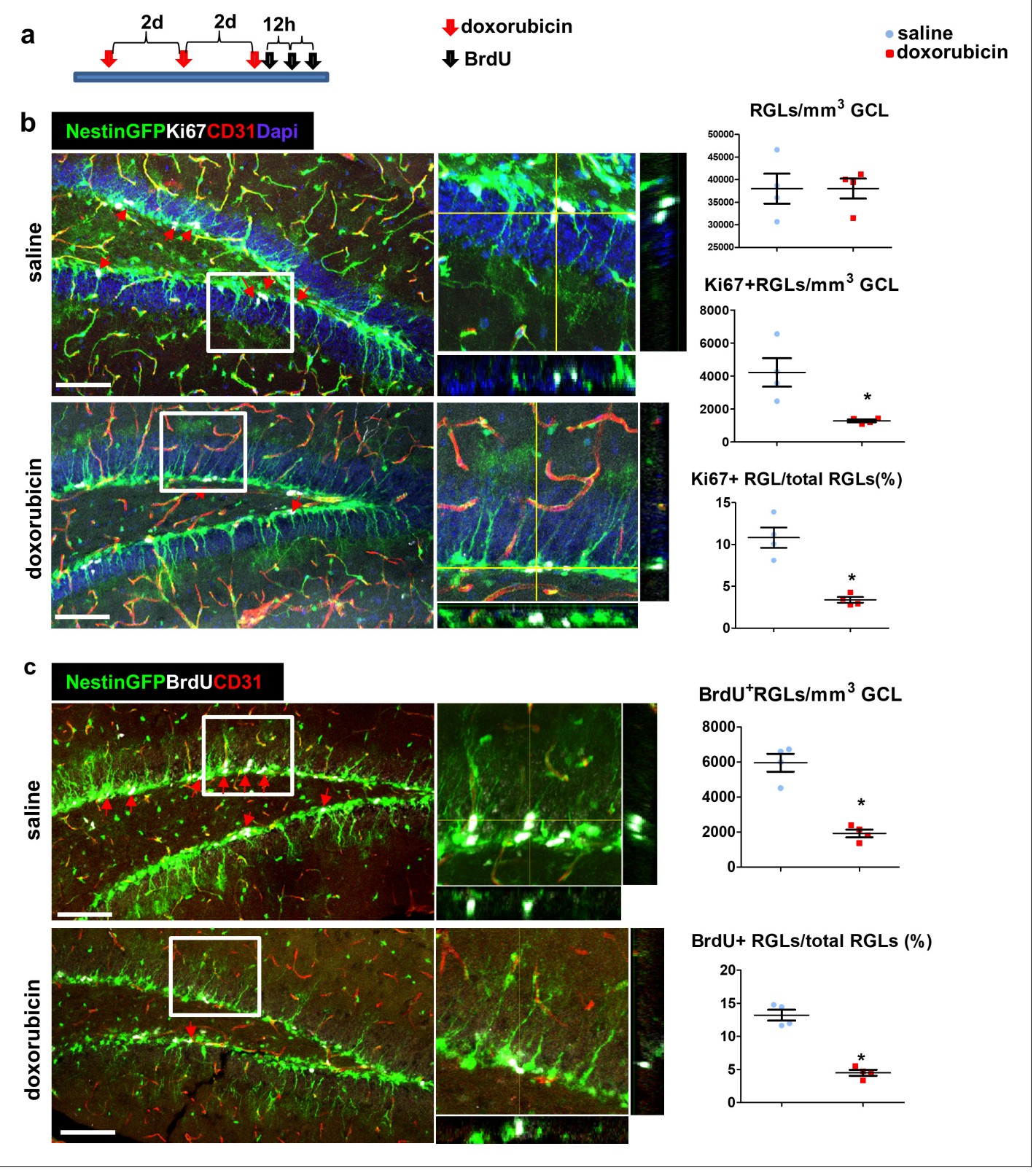

**Figure 8.** Doxorubicin treatment reduces the proliferation rates of Nestin-GFP⁺ RGLs. (a) Experimental protocol. Doxorubicin (5 mg/kg) or saline were intraperitoneally (I.P) injected to Nestin-GFP mice every other day (3 injections). BrdU (50 mg/kg) was injected I.P. to all animals 3 times at 12 hr intervals before brain retrieval. N = 4 in each group. (b) Immunofluorescence images of nestin-GFP DG co-stained with Ki67 (a proliferation marker) and CD31. Arrows highlight Ki67⁺GFP⁺ cells that were verified to have a radial cytoplasmic process (see enlarged z-projection insets). Right: quantification of the

*Figure 8 continued on next page*

Figure 8 continued

total Nestin-GFP RGL population density (top, t(6)=-0.016 p=0.988), Ki67$^+$ RGLs density (t(6)=3.398 p=0.015) and the percentage of Ki67$^+$ RGLs among total RGLs (t(6)=5.923 p=0.001). (c) DG sections were co-stained with BrdU and CD31. Arrows highlight BrdU$^+$GFP$^+$ cells that were verified to have a radial cytoplasmic process (see enlarged insets). Right: quantification of BrdU$^+$ RGLs density (t(6)=7.285 p=3.4*10$^{-4}$) and the percentage of BrdU$^+$ RGLs among total RGLs (t(6)=9.331 p=8.58*10$^{-5}$). Scale bar for all images, 100 µM.

ventricle 1–2 min before brain retrieval. For chronic use, doxorubicin was injected I.P. at 5 mg/kg every other day, three times.

## Tissue processing for immunofluorescence

Brains were fixed by perfusion and immersion in 4% PFA for 5 hr, incubated in 30% sucrose, embedded in OCT Tissue-Tec and cryosectioned to 50 µm floating sections. Coronal slices from all aspects of the rostral-caudal axis were examined. For BrdU staining, brain slices were incubated for 2 hr in 50% formamide/2xSSC at 65°C, 30 min in 2N HCl at 37°C and 10 min in 0.1M boric acid pH8.5. Staining was done as described (*Licht et al., 2010*) with the following: anti BrdU (Serotec 1:400 PRID:AB_323427), anti-CD31 (BD 1:50 PRID:AB_393571), anti Laminin (Thermo Scientific 1:200, RRID:AB_60396), Anti Collagen-IV (Abcam 1:200 PRID:AB_305584), Anti Ki67 (Thermo Scientific 1:200, PRID:AB_10979488), anti-GFP (Thermo Scientific 1:200, PRID:AB_221570) and anti-RFP (Abcam 1:400, PRID:AB_945213). Secondary antibodies were as described in Key Resource Table. Sections were mounted with Permafluor mounting medium (Thermo Scientific, TA-030-FM) with Dapi (Sigma, D9542).

Confocal microscopy was done using Olympus FV-1000 on 20X and X60 objectives and 1.46 µm distance between confocal z-slices. 3d reconstruction images were processed by Bitplane IMARIS 7.6.3 software using the Isosurface function.

## Cell proliferation quantification

6–7 confocal z-stack images per animal were quantified (N = 4 in every group). The volume of the GCL was calculated by measuring the GCL area in the image (using Olympus FV-10 viewer) and multiplying the measured area by the number of z-axis slices and the distance between slices (1.46 µm). Cells within the GCL (Ki67/CldU/GFP) were counted manually by a blind experimenter (using Olympus FV-10 viewer). Z-plane projections used to verify double labeling. Student's T-test (SPSS) was used to compare between groups, assuming a normal distribution and unequal variance.

## Super-resolution microscopy

We have used the dSTORM system which allows for approximately 20 nm resolution by using photo-switchable fluorophores. 50µm-thick *Gli1-Cre$^{ERT2}$*; Ai9 mouse brain slices were immunostained with anti-RFP (EnCor 1:300 PRID:AB_2572308), anti laminin (Abcam 1:200, PRID:AB_298179) and anti CD31 (Bio-Rad 1:50, PRID:AB_321653). Secondary antibodies used: donkey Anti Chick Alexa Fluor568, donkey Anti-rabbit Alexa Fluor488 and donkey anti hamster Alexa Fluor647, all from Jackson Immunoresearch. After staining, brain slice was mounted on Poly-D-lysine coated coverslip (1.5H). dSTORM imaging was performed in a freshly prepared imaging buffer that contained 50 mM Tris (pH 8.0), 10 mM NaCl and 10% (w/v) glucose with an oxygen-scavenging GLOX solution (0.5 mg/ml glucose oxidase (Sigma-Aldrich), 40 µg/ml catalase (Sigma-Aldrich), 10 mM cysteamine MEA (Sigma-Aldrich) and 1% β mercaptoethanol *Encinas et al., 2011*. A Nikon Ti-E inverted microscope was used. The N- STORM Nikon system was built on TIRF illumination using 1.49 × 100 oil immersing objective and ANDOR DU-897 camera. 488, 568 and 647 nm laser line were used for activation with cycle repeat of~5000 cycles. Nikon NIS Element software was used for collections the data; analysis was performed by ThunderSTORM ImageJ plugin (*Ovesný et al., 2014*). Images in 2D were Gaussian fit of each localization. dSTORM signal at the ROI (enclosing the entire vessel) was normalized to the length of vessel circumference. All groups were tested for normal distribution using the Kolmogorov-Smirnov test. One-way ANOVA with Tukey's post-hoc analysis was used to compare between groups.

## Tissue processing for electron microscopy

### For labeling RGLs

Brains of *Gli1-Cre^{ERT2}*; Ai9 mice (3 days post tamoxifen induction) were immersion-fixed by 4% para-formaldehyde (wt/vol); 0.1% glutaraldehyde (vol/vol) in 0.1M phosphate buffer; and kept at 4°C for 24 hr. The brain was then washed in PBS solution, and cut in 50 μm coronal sections with a vibratome (Leica). For the immunoperoxidase process, the tissue was washed 10 min in 3% $H_2O_2$ solution, blocked by two washes in 0.5% BSA and the sections were incubated overnight in the primary antibody (anti RFP, Abcam 1:400, PRID:AB_945213) in 0.1% BSA-PBS with shaking at 25°C. Next, sections were washed in PBS solution and incubated in the secondary antibody (anti-rabbit HRP, Vector MP-7451–50) shaking for 2 hr at 25°C. Sections were then washed in PBS solution and incubated in DAB/plus chromogen solution (Abcam ab103723) for 6–10 min, followed by subsequent washes in PBS solution (30 min).

### For systemic HRP imaging

10 mg of HRP (Sigma P8250) was dissolved in 0.4 ml of PBS and injected into the tail vein of an adult (p90) mouse. After 30 min of HRP circulation, the brain was dissected and fixed by immersion in a 0.1 M sodium-cacodylate-buffered mixture (5% glutaraldehyde and 4% PFA) for 1 hr at room temperature followed by 5 hr at 4°C. Following fixation, the tissue was washed overnight in 0.1 M sodium cacodylate buffer and then cut in 50-μm-thick free-floating sections using a vibratome (Leica). Sections were incubated in DAB/plus chromogen solution (Abcam ab103723) for 6–10 min, followed by subsequent washes in PBS solution (30 min).

### For double-labeling of HRP and RGLs

HRP was injected as above, processing was done as for labeling RGLs.

Stained brain slices were transferred back to the fixative solution, rinsed 4 times, 10 min each, in cacodylate buffer and post-fixed and stained with 1% osmium tetroxide, 1.5% potassium ferricyanide in 0.1M cacodylate buffer for 1 hr. Tissue was then washed 4 times in cacodylate buffer followed by dehydration in increasing concentrations of ethanol consisting of 30%, 50%, 70%, 80%, 90%, 95%, for 10 min each step followed by 100% anhydrous ethanol 3 times, 20 min each, and propylene oxide 2 times, 10 min each. Following dehydration, the tissue was infiltrated with increasing concentrations of Agar 100 resin in propylene oxide, consisting of 25, 50, 75, and 100% resin for 16 hr each step. The tissue was then embedded in fresh resin and let polymerize in an oven at 60°C for 48 hr. Embedded tissues in blocks were sectioned with a diamond knife on a Leica Reichert Ultracut S microtome and ultrathin sections (80 nm) were collected onto 200 Mesh, thin bar copper grids. The sections on grids were sequentially stained with Uranyl acetate for 5 min and Lead citrate for 2 min and viewed with Jeol JEM1400 Plus microscope equipped with Gatan camera.

## Quantification of basement membrane coverage and vesicular-like structure

55 EM images containing a blood vessel were used for quantification (N = 2 animals). Out of which, 50 were taken from the inner molecular layer and 5 were taken from the hilus area. The length of the blood vessel abluminal interface was measured by ImageJ (NIH). The length of BM (greyish line of ~50 nm thickness, clearly separating the endothelium from brain parenchyma (as in *Figure 2a*)) was measured at both RGL contact point and when no RGL was present.

A circular structure of 50–100 nm in the cytoplasm of an endothelial cell encompassing a single membrane was counted as a vesicle. We counted the numbers of these organelles per length unit (μm) of the defined abluminal aspect (RGL/no RGL, BM/no BM). All groups were tested for normal distribution using the Kolmogorov-Smirnov test. One-way ANOVA with Tukey's post-hoc analysis was used to compare between groups.

## Acknowledgements

We wish to thank Yuval Dor and Itzhak Dershowitz for helpful suggestions,and assistance, Yael Friedman and Eddie Berenshtein for TEM support. This work was supported by the European Research

Council (ERC(grant, project VASNICHE (Grant # 322692) and the Israel Science Foundation (grants # 1882/16 and # 2402/16) to ABZ.

## Additional information

### Funding

| Funder | Grant reference number | Author |
|---|---|---|
| H2020 European Research Council | 322692 | Tamar Licht<br>Myriam Grunewald<br>Saran Kumar<br>Eli Keshet |
| Israel Science Foundation | 1882/16 | Esther Sasson<br>Batia Bell<br>Ayal Ben-Zvi |
| Israel Science Foundation | 2402/16 | Esther Sasson<br>Batia Bell<br>Ayal Ben-Zvi |

The funders had no role in study design, data collection and interpretation, or the decision to submit the work for publication.

### Author contributions

Tamar Licht, Conceptualization, Data curation, Formal analysis, Supervision, Funding acquisition, Validation, Investigation, Visualization, Methodology, Writing - original draft, Project administration, Writing - review and editing; Esther Sasson, Batia Bell, Data curation, Formal analysis, Investigation, Methodology; Myriam Grunewald, Saran Kumar, Data curation, Methodology; Tirzah Kreisel, Data curation, Formal analysis, Supervision, Dr. Kreisel performed statistical analysis of the new data in the revised manuscript. She accepts her position in the author list and approved the manuscript; Ayal Ben-Zvi, Data curation, Supervision, Investigation, Methodology, Project administration, Writing - review and editing; Eli Keshet, Formal analysis, Supervision, Investigation, Methodology, Project administration, Writing - review and editing

### Author ORCIDs

Tamar Licht (iD) https://orcid.org/0000-0003-2333-1665

### Ethics

Animal experimentation: All animal procedures were approved by the animal care and use committee of the Hebrew University (Protocol MD-14-13900-3). Every effort was made to minimize the numbers of animals and suffering.

### Decision letter and Author response

Decision letter https://doi.org/10.7554/eLife.52134.sa1
Author response https://doi.org/10.7554/eLife.52134.sa2

## Additional files

### Supplementary files

• Transparent reporting form

### Data availability

All data generated or analysed during this study are included in the manuscript.

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
