## [Decision Letter]

Thank you for submitting your article "Hippocampal neural stem cells facilitate access from circulation via apical cytoplasmic processes" for consideration by *eLife*. Your article has been reviewed by three peer reviewers, and the evaluation has been overseen by Kate Wassum as the Senior and Reviewing Editor. The following individuals involved in review of your submission have agreed to reveal their identity: Chenghua Gu (Reviewer #1); Christer Betsholtz (Reviewer #2).

The reviewers have discussed the reviews with one another and the Reviewing Editor has drafted this decision to help you prepare a revised submission.

The data in this report demonstrate a unique and novel type of connection between neural stem cells of the brain's dentate gyrus and vascular endothelial cells of capillaries that feed this area. The data show that these contact sites function to actively deliver blood-borne molecules to the dentate gyrus cells. The chemotherapeutic agent doxorubicin is shown to be taken up specifically by dentate gyrus cells, and this leads to their reduced proliferation. These data, thus, explain why doxorubicin has cognitive side effects in spite of the fact that it should not pass the normal blood-brain barrier. These are significant novel discoveries of pathophysiological and physiological relevance. This manuscript opens new understanding of how systemic factors regulate dentate gyrus cell function in wanted and unwanted ways.

The reviewers noted the following concerns that must be thoroughly addressed.

1) The increase of transcytotic vesicles at the touching points of endothelial cells and neural stem cells is not convincingly shown. A tracer that is detectable by EM, such as HRP, should be injected into circulation to demonstrate tracer uptake by transcytotic vesicles and neural stem cells. The transcytotic vesicles in endothelial cells in touch or not in touch with neural stem cells should be quantified. EM quality should be improved.

2) The authors should quantify vesicle density in Figure 3 and confirm that these vesicles are related to transcytosis (clathrin or caveolar), and not incidental (e.g., lysosomes, Golgi vesicles).

3) Basement membrane coverage of endothelial cells in touch or not in touch with neural stem cells needs to be quantified. The authors should quantify their TEM images to strengthen their claims about a thinned BM at the BV-RGL interface. The absence of a BM is not clear in Figure 2B, especially in the top-left image where a BM appears to continuously move upwards even at the BV-RGL interface, and in Figure 3. Moreover, Figure 2I-N of prior work from Moss, et al., 2016, appear to show a BM at the BV-RGL interface.

4) Is the BM similarly thinned at BV areas covered by pericytes and astrocytes-or is this specific to RGLs? And is a BM similarly lacking in the BV-NSC interface in the SVZ? This would support specificity of BM thinning at the BV-RGL interface and thus claims regarding its role in the uptake of circulatory factors.

5) The authors suggest that the BV-RGL interface is a "fully functional BBB" but that rapid uptake of injected dextrans and doxorubicin is mediated by a thinned BM at the BV-RGL interface. The literature would suggest that a fully functional BBB precludes the uptake of such factors by the brain endothelial cells, upstream of access to the BM. This has been found with dextrans, HRP, and endogenous plasma proteins. Please address this discrepancy. For example, the authors could replicate established experiments on these particular BVs to determine if they are indeed fully functional, such as looking for HRP+ vesicles in endothelial cells by TEM (e.g., Ben-Zvi et al., 2014, Figure 5) and staining for endogenous plasma protein uptake (e.g., PMID 28441414, Figure 6). Relatedly, functional brain BVs are also marked by "extremely low rates of transcellular vesicular transport" (Ben-Zvi et al., 2014). The authors are asked to reconcile this with their findings in Figure 3.

6) The authors should tone down their conclusion or provide direct evidence for their claim that doxorubicin uptake by RGLs reduces their proliferation: do doxorubicin+ RGLs display less Ki67 and BrdU signal compared to doxorubicin- RGLs? Because doxorubicin fluoresces red, this should be visible in existing tissue slices from Figure 7B.

[Editors' note: further revisions were suggested prior to acceptance, as described below.]

Thank you for resubmitting your work entitled "Hippocampal neural stem cells facilitate access from circulation via apical cytoplasmic processes" for further consideration by *eLife*. Your revised article has been reviewed by two peer reviewers and the evaluation has been overseen by Kate Wassum as the Senior and Reviewing Editor.

The manuscript has been improved but there are some remaining issues that need to be addressed before acceptance, as outlined below:

1) The newly introduced data in Figure 3 has been quantified by comparing blood vessels in the DG with blood vessels in cortex. It is unclear whether there are general differences of basement membrane coverage within different brain regions. Therefore, and also for consistency (with quantifications in Figures 2C/4D and E) it would be preferential to do the quantification by comparing BV-RGL contact points with non-contact points of blood vessels within the DG.

2) Figure 2B top left panel shows a continuous basement membrane between EC and RGL. To be more representative of the quantified data and to better reflect the new finding, it would be beneficial to replace this image with an image showing a discontinued basement membrane at the RGL contact point.

3) It is difficult to find orientation in Figure 4F top right panel. Maybe labeling of lumen and basement membrane would help. Ideally this image could be replace with an image that meets the quality of the other panels in this figure.

4) It is unfortunate that the authors could not quantify tracer filled vesicles at RGL-BV contact points in comparison with non-contact points. This appears to be a crucial piece of data (as it was also demanded by more than one reviewer) to verify increased transcytotic activity (in a spatially very confined region). Would a double labeling of RGLs and HRP by DAB be possible? The labeled structures appear in different cells and are morphologically distinguishable. Alternatively, could this quantification be done by determining the region based on the absence of basement membrane?

---

## [Author Response]

[…] The reviewers noted the following concerns that must be thoroughly addressed.1) The increase of transcytotic vesicles at the touching points of endothelial cells and neural stem cells is not convincingly shown. A tracer that is detectable by EM, such as HRP, should be injected into circulation to demonstrate tracer uptake by transcytotic vesicles and neural stem cells. The transcytotic vesicles in endothelial cells in touch or not in touch with neural stem cells should be quantified. EM quality should be improved.2) The authors should quantify vesicle density in Figure 3 and confirm that these vesicles are related to transcytosis (clathrin or caveolar), and not incidental (e.g., lysosomes, Golgi vesicles).

We performed experiments with tail-vein injections of HRP and TEM imaging. These are presented in new Figure 4. We were able to demonstrate cytoplasmic and abluminal HRP-filled vesicles. These detection methods, however, are not sensitive enough to label HRP accumulation in non-endothelial cells. Additionally, since the same specimen cannot be labeled with DAB to both injected HRP and to RFP in stem cells, we could not pinpoint the specific location of BV-NSC contact in HRP-injected specimens.

In RFP-labeled TEM images, we quantified the density of vesicular-like structures at contact points (in comparison with areas with no NSC contact) and found a significantly higher density. These vesicular-like structures were determined as circular organelles of 60-100nm. We also quantified the densities of these structures in NSC-BV contact points with or without evident basement membrane and found a significantly higher number in areas without evident BM. All of these results appear in Figure 4.

3) Basement membrane coverage of endothelial cells in touch or not in touch with neural stem cells needs to be quantified. The authors should quantify their TEM images to strengthen their claims about a thinned BM at the BV-RGL interface. The absence of a BM is not clear in Figure 2B, especially in the top-left image where a BM appears to continuously move upwards even at the BV-RGL interface, and in Figure 3. Moreover, Figure 2I-N of prior work from Moss, et al., 2016, appear to show a BM at the BV-RGL interface.

As suggested, we have quantified BM coverage at NSC-BV contact points in comparison with BV areas devoid of NSC, or in the hilus where no NSC processes exist. We found that in BV with no contacts, the majority of the abluminal aspect is surrounded by evident BM. At contact points, however, some areas do have a BM but the majority have not. These quantifications are presented in new Figure 2. Indeed, since these BM-reduced contacts are seemingly formed at specific points and not all through the NSC arm, Moss et al. presented an example of contacts that include a basement membrane (as we also do, see Figure 2C).

Figure 2B (left) was enlarged to better highlight the BM. The new image in Figure 4A better shows reduced BM at the contact points with two RGLs.

We added confocal and super-resolution microscopy images to further establish the finding of reduced BM components at contact points. New Figure 3 shows laminin immunostaining and uses both high magnification confocal images and STORM microscopy. New Figure 3—figure supplement 1 shows reduced Collagen IV staining at contact points.

4) Is the BM similarly thinned at BV areas covered by pericytes and astrocytes-or is this specific to RGLs? And is a BM similarly lacking in the BV-NSC interface in the SVZ? This would support specificity of BM thinning at the BV-RGL interface and thus claims regarding its role in the uptake of circulatory factors.

We thank the reviewer for both of these suggestion. We have indicated in the revised manuscript (Figure 2A legends) that the BM is not thinned in contact points with pericytes and astrocytes. The pericytic contact point is characterized by a double layer of BM surrounding both the EC and the pericyte. In astrocytic endfeet (that accompany the majority of cerebral capillaries), the BM is clearly evident. Example for both is seen in Figure 2A.

We were unable to characterize the SVZ NSC-BV contact points in TEM as these contacts are rare and localized to the medial striatum near the lateral ventricles. However, we could neither identify the 10Kd tracer accumulation in SVZ NSC (see Figure 5A), therefore we believe that the case of the SVZ could be different. SVZ stem cells are also in contact with the CSF in ventricles (Mirzadeh et al., 2008) and most likely receive inputs by this route.

5) The authors suggest that the BV-RGL interface is a "fully functional BBB" but that rapid uptake of injected dextrans and doxorubicin is mediated by a thinned BM at the BV-RGL interface. The literature would suggest that a fully functional BBB precludes the uptake of such factors by the brain endothelial cells, upstream of access to the BM. This has been found with dextrans, HRP, and endogenous plasma proteins. Please address this discrepancy. For example, the authors could replicate established experiments on these particular BVs to determine if they are indeed fully functional, such as looking for HRP+ vesicles in endothelial cells by TEM (e.g., Ben-Zvi et al., 2014, Figure 5) and staining for endogenous plasma protein uptake (e.g., PMID 28441414, Figure 6). Relatedly, functional brain BVs are also marked by "extremely low rates of transcellular vesicular transport" (Ben-Zvi et al., 2014). The authors are asked to reconcile this with their findings in Figure 3.

We understand that the terminology we used might not be accurate enough. We believe that BV-RGL contact points are characterized not only by thinned BM (Figure 2) but also by a higher vesicular activity (Figure 4). Therefore, while the majority of endothelial cells in the region indeed exhibit fully functional BBB properties, endothelial cells in contact with RGL (or even only sub-cellular sections of these cells) facilitate ‘non-barrier’ functions. In Figure 4, we now present HRP+ vesicles in endothelial cells. The density of vesicles at endothelial cells of the inner molecular layer is much higher than those published for capillaries elsewhere in the brain (e.g. Ben-Zvi et al., 2014).

Relevant text related to this issue was changed to better illustrate the exact hypothesis. We highlighted in the last two paragraphs of Introduction and the first paragraph of the Discussion that the hippocampus is thought to have a fully functioning BBB milieu while we found a sub-region in which the BBB is altered.

6) The authors should tone down their conclusion or provide direct evidence for their claim that doxorubicin uptake by RGLs reduces their proliferation: do doxorubicin+ RGLs display less Ki67 and BrdU signal compared to doxorubicin- RGLs? Because doxorubicin fluoresces red, this should be visible in existing tissue slices from Figure 7B.

This is indeed a correct observation However, the protocol for doxorubicin labeling of stem cells is an acute intracardial injection. Since the chest is open, the mouse is sacrificed within 2 minutes and therefore the effect on stem cell proliferation cannot be measured. In contrast, the long-term doxorubicin exposure in the second setting used for NSC proliferation inhibition is three IP injections of doxorubicin, every other day. The doses reaching the brain in this protocol are not sufficient to be detected by fluorescence. We believe that doxorubicin accumulates equally in all NSC but affect only those who are engaged in proliferation. We added a text explaining this technical difference in dox protocols in the Results section. We also toned down this phrase in the Discussion and explained that dox potentially inhibit cell proliferation due to its penetrance to stem cells and maybe not solely by secondary mediators.

[Editors' note: further revisions were suggested prior to acceptance, as described below.]

The manuscript has been improved but there are some remaining issues that need to be addressed before acceptance, as outlined below:1) The newly introduced data in Figure 3 has been quantified by comparing blood vessels in the DG with blood vessels in cortex. It is unclear whether there are general differences of basement membrane coverage within different brain regions. Therefore, and also for consistency (with quantifications in Figures 2C/4 and E) it would be preferential to do the quantification by comparing BV-RGL contact points with non-contact points of blood vessels within the DG.

As suggested, for the revised manuscript, we expanded the dSTORM-based analysis of blood vessels within the DG to include comparison between vessels with RGL association and vessels in the vicinity of RGLs with no RGL contacts. The data are presented in new Figure 3H-N, thereby also increasing the numbers of RGL-associated capillaries analyzed.

2) Figure 2B top left panel shows a continuous basement membrane between EC and RGL. To be more representative of the quantified data and to better reflect the new finding, it would be beneficial to replace this image with an image showing a discontinued basement membrane at the RGL contact point.

We replaced this image to clearly demonstrate discontinued BM at the contact point with DAB-labeled RGL.

3) It is difficult to find orientation in Figure 4F top right panel. Maybe labeling of lumen and basement membrane would help. Ideally this image could be replace with an image that meets the quality of the other panels in this figure.

As suggested, we replaced this image with a better quality image and also added labels to better explain the orientation. The new image includes examples of HRP-filled vesicles near a vague BM structure (presented in Figure 4E). For comparison, we included a representative image of HRP-filledcapillary lumen aligned with an astrocyte end-feet with a clear BM and no vesicular activity (Figure 4F).

4) It is unfortunate that the authors could not quantify tracer filled vesicles at RGL-BV contact points in comparison with non-contact points. This appears to be a crucial piece of data (as it was also demanded by more than one reviewer) to verify increased transcytotic activity (in a spatially very confined region). Would a double labeling of RGLs and HRP by DAB be possible? The labeled structures appear in different cells and are morphologically distinguishable. Alternatively, could this quantification be done by determining the region based on the absence of basement membrane?

To address this important point, we followed the reviewer's suggestion to try a novel approach of double labeling for both RGLs termini and transcytotic endothelial vesicles with DAB (Immuno-EM with anti-RFP and secondary HRP conjugated antibody for RGL and HRP injected as tracer for vesicles). We have to admit that we found this approach technically challenging and not a trivial combination of the two protocols: Using Immuno-EM requires a very low fixative concentration, not ideal for achieving high-quality EM images. Moreover, immunostaining brain slices with anti-RFP antibody involves washing steps which interferes with the detection of HRP in BVs lumen. Nevertheless, we preformed the requested quantification based on HRP traces at the endothelial luminal surface and in vesicles. While not ideal, we believe the data indeed support the notion of increased transcytotic activity. These results are presented in Figure 4—figure supplement 1B-D.